# Isolation of the side population from neurogenic niches enriches for endothelial cells

**Alena Kalinina[1], Catherine Gnyra[1], Vera Tang[2], Yingben Xue[1], Diane C. Lagace[1]***

**1** Department of Cellular and Molecular Medicine, Neuroscience Program, Ottawa Hospital Research Institute, Brain and Mind Research Institute, University of Ottawa, Ottawa, Canada, **2** University of Ottawa Flow Cytometry and Virometry Core Facility, Ottawa, Ontario, Canada

* diane.lagace@uottawa.ca

**Data Availability Statement:** All relevant data are within the paper and its Supporting information files.

**Funding:** Specific grant numbers: RGPIN-2020-06541 Initials of authors who received each award

## Abstract

In stem cell research, DNA-binding dyes offer the ability to purify live stem cells using flow cytometry as they form a low-fluorescence side population due to the activity of ABC transporters. Adult neural stem cells exist within the lateral ventricle and dentate gyrus of the adult brain yet the ability of DNA-binding dyes to identify these adult stem cells as side populations remains untested. The following experiments utilize the efflux of a DNA-binding dye, Vyrbant DyeCycle Violet (DCV), to isolate *bona fide* side populations in the mouse dentate gyrus and subventricular zone (SVZ), and test their sensitivity to ABC transporter inhibitors. A distinct side population was found in both the adult lateral ventricle and dentate gyrus using DCV fluorescence and forward scatter instead of the conventional dual fluorescence approach. These side populations responded strongly to inhibition with the ABC transporter antagonists, verapamil and fumitremorgin C. The majority of the cells residing in the side populations of dentate gyrus and SVZ were characterized by their expression of CD31. Additionally, at least 90% of all CD31+ cells found in the dentate gyrus and SVZ were negative for the hematopoietic marker CD45, leading to the hypothesis that the CD31+ cells in the side population were endothelial cells. These findings, therefore, suggest that the side population analysis provides an efficient method to purify CD31-expressing endothelial cells, but not adult neural stem cells.

## Introduction

DNA-binding dyes have been perpetually used in flow cytometry and fluorescence-activated cell-sorting (FACS) paradigms to identify cancer stem cells [1, 2]. This has included the use of dyes such as Hoechst 33342 [3, 4] and, more recently, Vybrant DyeCycleViolet (DCV), which is less toxic to stem cells [5–7]. In these assays, live stem cells are identified as a side population that has low dual fluorescence intensity in both blue- and red-shifted spectra due to the activity of ABC transporters, which can efflux the DNA-binding dyes. In contrast, cells that do not have ABC transporters will accumulate the dye and show higher fluorescence. Since ABC

DL Full names of commercial companies that funded the study or authors None Initials of authors who received salary or other funding from commercial companies None URLs to sponsors' websites None The funders had no role in study design, data collection and analysis, decision to publish, or preparation of the manuscript.

**Competing interests:** The authors have declared that no competing interests exist.

transporters have been identified in stem cells from a large variety of tissues [8], this method has extended to be used routinely to isolate various types of stem cells.

Neural stem cells (NSCs) within the subventricular zone (SVZ) of the lateral ventricle and the subgranular zone of the dentate gyrus can develop into functional mature neurons in the adult brain. There is interest in harvesting cells from these regions in order to understand how NSCs and their progeny contribute to brain function in health and disease and could be harnessed for cell-based brain repair [9, 10]. The isolation of neural stem cells (NSCs) has been a challenge in the field due to many reasons. These include the relatively small numbers of NSCs, the multiple subtypes of NSCs, and lack of a highly specific surface marker for identification and isolation by FACS [9, 11]. Classically, flow cytometry paradigms utilized low expression of PNA (peanut agglutinin) and HAS (heat stable antigen), or high expression of Notch, LewisX (cluster of differentiation-15, CD15) and EGFR (early growth response factor) surface markers to identify neural stem cells [12–15]. More recent methods have made additional significant advances in identification and enrichment of subpopulations of NSCs using multi-parameter FACS, inducible transgenic mice models [13], or single-cell transcriptional analyses [16–19]. However, these methodologies are time- and cost-intensive, which has led our lab and others to investigate the use of DNA-binding dyes as a simpler and more efficient method for identification and purification of NSCs.

Many have identified an ABC transporter-dependent side population with an NSC identity in cells isolated from neurospheres that were derived from primary embryonic neural or postnatal/adult SVZ tissue [3, 12, 20]. In contrast, NSCs isolated *ex vivo* in cells freshly harvested from embryonic or early postnatal SVZ (postnatal day 2) brain tissue are not found in the side population [3, 12, 20]. Instead of NSCs, endothelial and microglial cells were comprising the side population identified in *ex vivo* preparations of developing SVZ [3]. This finding is not surprising as, endothelial and microglial cells along with pericytes and astrocytes, form and maintain the blood brain barrier [21–23]. Accordingly, one of the main roles of endothelial cells is in brain homeostasis, which relies on the function of the ABC transporters [23].

This raises the question of whether NSC-containing side populations can be identified from *ex vivo* primary adult mouse dentate gyrus and SVZ tissue. To answer this question, we optimized the detection and phenotyping of the side population using flow cytometry and the DNA-binding dye, DCV, in live single-cell suspensions from the young adult mouse dentate gyrus and SVZ. The data shows that an ABCG2/B1-dependent side population can be identified in the neurogenic niches that is enriched for CD31-expressing endothelial cells but not NSCs.

## Materials and methods

### Animals

This study was carried out in strict accordance with the recommendations in the Guidelines of the Canadian Council on Animal Care and all efforts were made to minimize suffering. The animal care protocol was approval by the University of Ottawa Animal Care Committee (Protocol CMM-1150). Fifty-six male and female two to three months old C57bl/6J background mice were used for all experiments. Animals were group housed in standard laboratory cages and kept on a 12-hour night/day cycle with *ad libitum* access to food and water.

### Tissue collection and digestion

Mice were deeply anesthetized with euthanyl (90 mg/kg) and the brains were quickly placed in ice-cold artificial cerebrospinal fluid (aCSF, pH = 7.4) prepared in miliQ water with 124mM NaCl, 5mM KCl, 1.3 mM MgCl2·6H2O, 2mM CaCl2·2H2O, 26mM NaHCO3, and 1X

penicillin-streptomycin (10,000 U/mL; ThermoFisher) and sterilized using stericup and steri-top filtration set (Millipore). Dentate gyrus and SVZ were microdissected using SteREO Discovery V8 microscope (Zeiss) following previously published protocols [24, 25].

Tissue was digested according to protocols described previously [26, 27]. First, the tissue was gently broken up using small surgical scissors then incubated on shaker (30 minutes, 37°C) in 500uL of digestion media, containing 20 U/mL papain (Worthington Biochemicals), 12 mM EDTA (Invitrogen) in DMEM:F12 (Invitrogen). Following incubation, Resuspension media (0.05 mg/mL DNase1 (Roche) with 10% fetal bovine serum (Wisent Bioproducts) in DMEM:F12) was added to each tube, triturated 10X with a P1000 micropipette, and incubated for five minutes at RT. Suspension was then transferred in Percoll media, consisting of 19.8% Percoll (GE Healthcare Life Sciences), 2.2% 10X PBS (Wisent Bioproducts) in Resuspension media. Cells were then spun down (500 x g, 13 minutes, 4°C), the supernatant was removed. For each experiment, cells from multiple mice were pooled into one dentate gyrus sample and one SVZ sample, and were resuspended in phenol-free DMEM:F12. Live cells were counted on Countess automated cell counter (ThermoFisher Scientific) using 0.4% Trypan blue (Invitrogen) at a concentration of 1:2 and suspended in phenol-free DMEM:F12 medium at a concentration of $10^6$/mL.

## Staining and drug treatments

To generate negative, single-stained, and all-stained samples, an average of eight mice was used per experiment. After splitting cells based on staining conditions, Vybrant DyeCycle Violet Ready Flow™ Reagent (Invitrogen) was added to cells in phenol-free DMEM:F12 medium and incubated at 37°C in a 5% CO2 cell culture chamber (Forma Series II Water Jacket; ThermoFisher Scientific) for 30 minutes. The concentration of DCV was tested at both 1X and 2X, and based on these experiments (S1 Fig), all future experiments used the concentration of 2X, or 160uL in $10^6$ cells/ml.

For experiments involving ABC transporter inhibition, fumitremorgin C (FTC; Sigma) and (±) verapamil hydrochloride (VP; Sigma) were added to unwashed cells at final concentrations of 10uM and 50uM, respectively, after DCV incubation and kept in the same conditions for additional 30 minutes. Cells were then kept on ice in dark until sort, and 7-Amino-Actinomycin D (7AAD, 40 ug/ml, Sigma) was added to cell suspensions 10 minutes before analysis for dead cell discrimination. For experiments determining the identity of the side population, CD31 antibody conjugated to allophycocyanin (APC), BD Biosciences, BioLegend), was added to cells in DMEM:F12 at final concentration of 1:50 [28] and incubated on ice in the dark for 30 minutes before DCV incubation, which followed the same workflow as discussed above. For supplementary experiment, CD45 antibody conjugated to fluorescein iso-thiocyanate (FITC) was co-incubated together with APC anti-CD31 on ice at a concentration of 1:500 for 30 minutes. Antibodies, dyes, and drugs used for all experiments are listed in Table 1.

## Cell lines

Two cancer cell lines, U-2OS (ATCC, osteosarcoma) and A2780 S ([29], ovarian cancer), were generously provided by Dr. Laura Trickle-Mulcahy and Dr. Barbara Vanderhyden, respectively. Cells were grown in DMEM/10%FBS until minimum 75% confluency was reached. Cells were detached from flasks in 5mM EDTA for 20 minutes at 37°C in a cell culture incubator, then triturated and washed several times with 1X PBS before cell count and DCV staining, which followed the same procedure as staining in primary brain cells.

**Table 1. Reagents used for tissue processing and DCV assay.**

| Reagents | Company | Catalogue # | Final Concentration |
|---|---|---|---|
| Vybrant DyeCycle Violet Ready Flow™ Reagent | Invitrogen | R37172 | 160 uL/ml |
| APC anti-mouse CD31 | BioLegend | 102409 | 1:50 |
| FITC anti-mouse CD45 | BioLegend | 147709 | 1:500 |
| 7AAD | Sigma | A9400-1MG | 1 ug/ml |
| Fumitremorgin C | Sigma | F9054-250UG | 10uM |
| (+/-) Verapamil Hydrochloride | Sigma | V4629-1G | 50uM |
| Papain suspension | Cedarlane | LS003126 | 20 U/ml |
| Percoll | Sigma | 17-0891-02 | 22% v/v |
| Trypan Blue | Invitrogen | T10282 | 1:1 (0.2%) |

## Flow cytometry

All flow cytometry experiments were performed using BD LSRFortessa™ flow cytometer (BD Biosciences) in the Flow Cytometry and Virometry Core at the University of Ottawa, Faculty of Medicine. Unstained and single-stained controls were used to set up laser parameters and gating for all-stained samples. First, cell debris and doublets were excluded based on FSC and SSC parameters, and then 7AAD+ dead cells were removed from analyses. Following this, all samples were collected under 405nm laser with 450/50 and 660/20 bandpass filters. DCV+ populations could only be resolved with optimal excitation of the samples (S1 Fig). 7AAD signal was collected under the 561nm laser with a 670/30 filter. APC-CD31 fluorescence was collected using the 640nm laser with a 660/20 bandpass filter without compensation. FITC-CD45 signal was collected using the 488 laser with a 530–30 bandpass filter without compensation. Single-stained controls were used to identify and gate CD31+ and CD31- cells. The side population fidelity of DCV+ cells was determined by comparison to FTC- and VP-treated samples. The number of live single cells analyzed in all-stained samples averaged 160±20k live single cells for all experiments, with the full entirety of the samples not run.

## Data analysis

FlowJo software (BD Biosciences) and GraphPad Prism 8 (GraphPad Software) were used to analyze and visualize all flow cytometry data. All average values are reported as mean ± standard error. All relevant data are within the manuscript and its Supporting files.

## Results

### Primary cells isolated from the dentate gyrus and SVZ contain multiple populations with side population properties

We used the Vybrant DyeCycle Violet Ready Flow Reagent™ (Invitrogen) to test the presence of a side population that was able to efflux the DNA-dye. Primary live cells harvested from the dissected neurogenic regions of the dentate gyrus and SVZ showed heterogeneous populations of DCV-stained cells as demonstrated by variable DNA content (Fig 1A and 1B). In both the dentate gyrus and SVZ populations there was a large population of cells with low DCV fluorescence that appeared in the lower left corner of dual fluorescence DCV-Blue/DCV-Red plots (Fig 1A and 1B). These cells in the lower corner resembled effluxing cells, which were absent in the negative control U2OS cell line (Fig 1C) that has been previously reported to not contain a side population [30, 31].

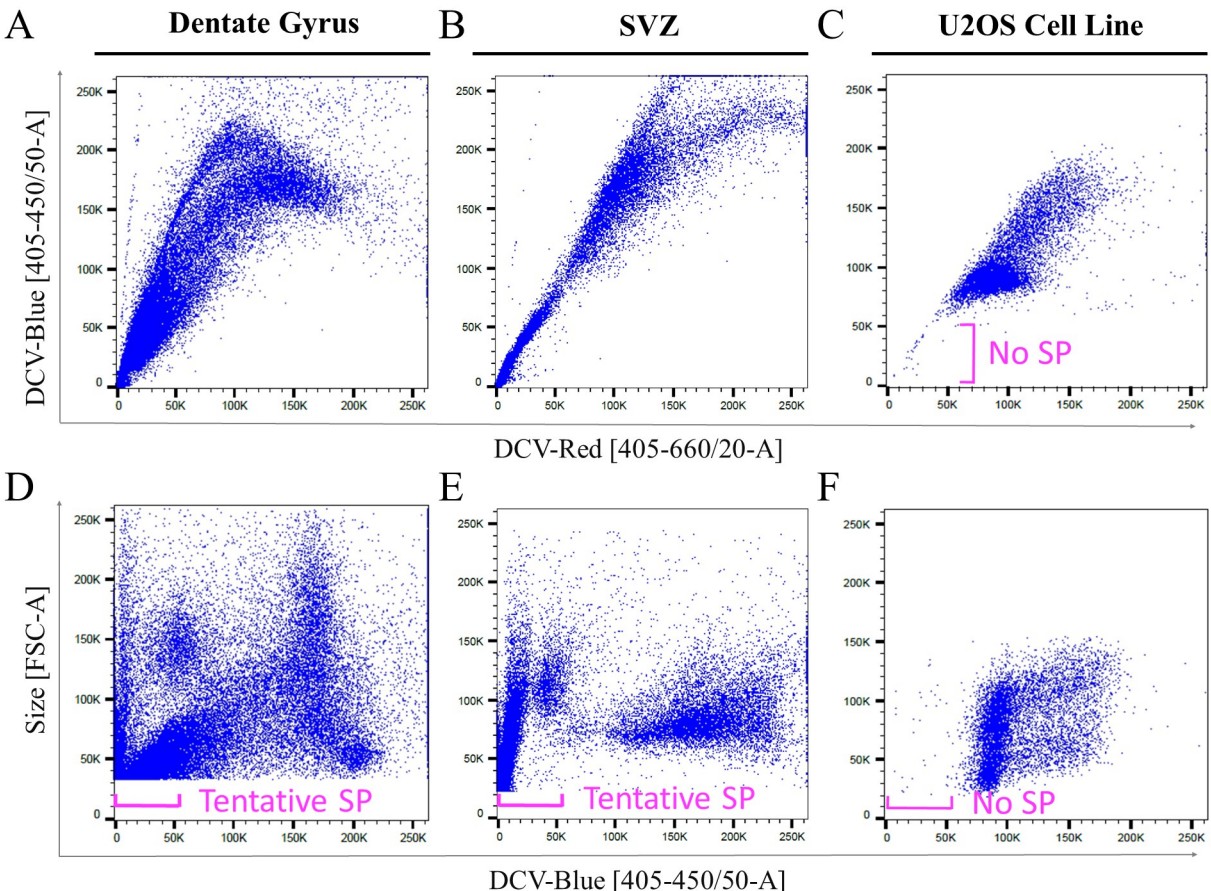

**Fig 1. DCV identifies heterogeneous cell populations in primary neurogenic brain cells.** Cell heterogeneity is illustrated in dual fluorescence DCV-Red vs DCV-Blue plots with adult cells isolated from the dentate gyrus (A), SVZ (B), as well as control cultured U2OS cells (C) that do not have a side population. Forward scatter (size) and DCV-Blue combined plots for dentate gyrus (D) and SVZ (E) show multiple low-fluorescence populations that could be bona fide side populations (labeled as Tentative SP), whereas, U2OS cells (F) show a few scattered cells that are debris and nuclear fragments. Plots A and D are representative plots based on samples pooled from two male and ten female mice. Plots B and E are representative plots based on samples pooled from two male and seven female mice.

In the cell suspensions from the dentate gyrus and SVZ it was difficult to distinguish the side population from the continuous main population using the red and blue dual fluorescence of DCV. Therefore, a forward scatter parameter (cell size) was added for the remainder of our analyses to observe the cell heterogeneity together with DCV fluorescence (Fig 1D–1F). The size/DCV-Blue plots of dentate and SVZ cells reveal multiple low-fluorescence cell populations that appeared to be effluxing DCV, which we labeled as the tentative side population (Fig 1D and 1E). As expected the DCV-stained live U2OS negative control cells showed no low-fluorescence cell populations (Fig 1F).

## Side populations in primary dentate gyrus and SVZ are responsive to ABC transporter inhibition

To further identify the side population of the adult neural cells, fumitremorgin C (FTC) and verapamil (VP) were used to inhibit the activity of ABCG2 and ABCB1 transporters, which are known to prevent the efflux of DCV [5, 32]. Primary dentate gyrus cells located in the labeled tentative SP area in the size-DCV-blue plot, showed a population of cells that was reduced

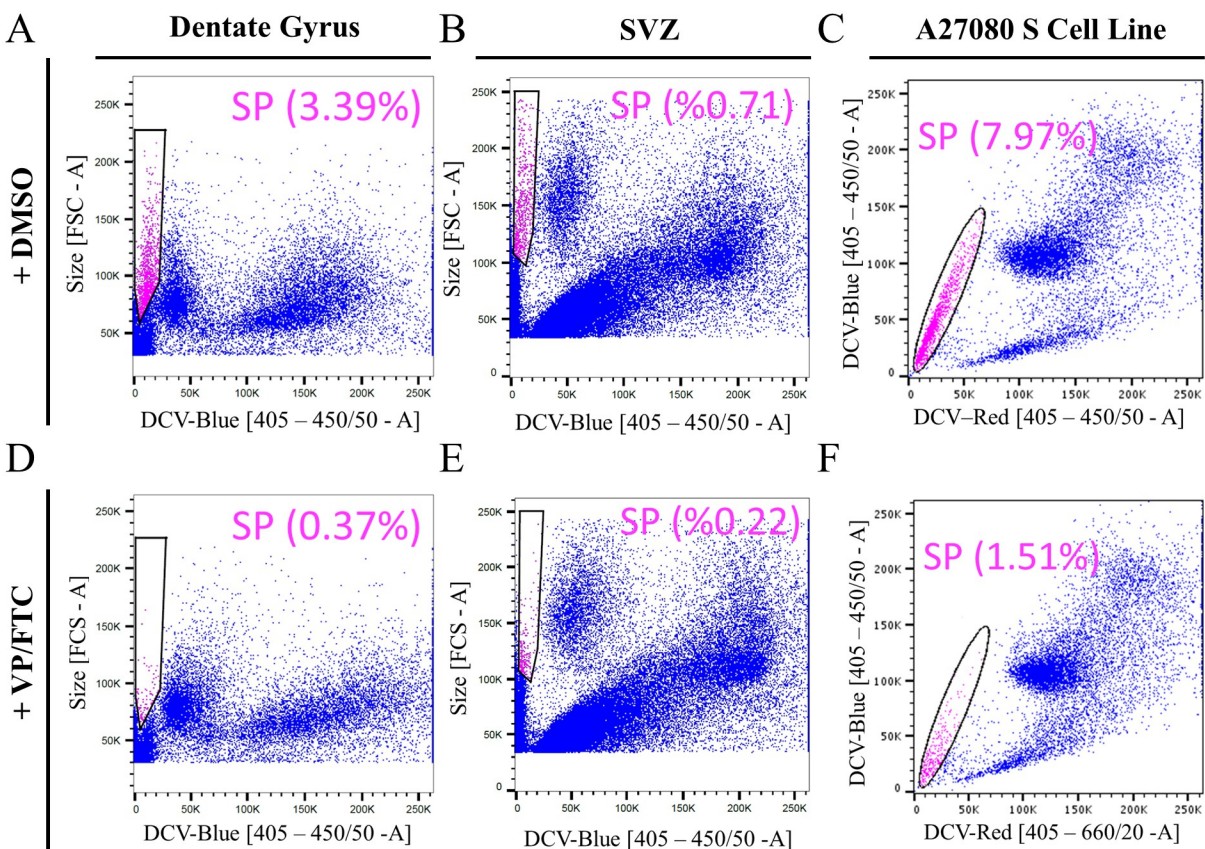

**Fig 2. ABC transporter antagonists (VP and FTC) inhibit the side population phenotype of primary adult dentate gyrus and SVZ cells.** Fluorescence/size plots for dentate gyrus (A) and SVZ (B) show the side population (SP, pink) and percentage of cells in the SP in the absence (A, B) and presence of VP and FTC (D, E) which significantly reduced the side population. A2780 S cell line, a positive control, shows a large side population in dual fluorescence plots (C) and nearly complete removal (F) of the effluxing side population by addition of inhibitors. Plots A and D are representative plots based on samples pooled from five male and three female mice. Plots B and E are representative plots based on samples pooled from six female mice.

from 3.39% to 0.37% after addition of ABC-transporter antagonists (Fig 2A and 2D). Similarly, primary SVZ cells located in the tentative SP area responded to the inhibitor treatment with a reduction of the population from 0.71% to 0.22% after treatment (Fig 2B and 2E). To confirm specificity of ABC transporter inhibitors, the A2780 S cell line was used as a positive control for VP- and FTC-sensitive side population cells [2, 7, 33]. As predicted, A2780 S cells had a very distinct side population in standard dual-fluorescence plots that was reduced from 7.97% to 1.51% after treatment (Fig 2C and 2F).

Examination of all performed experiments revealed that the size of dentate gyrus and SVZ side populations was reproducible between experimental days. Specifically, the average size of the dentate gyrus side population from 5 experiments was 3.82±0.26% of all live singlets, with 4050±1099 SP cells analyzed per sample. Similarly, the average size of the SVZ side population from 3 experiments was SVZ of 1.48±0.47% of all live singlets and contained 3328±1817 live single cells per sample. Overall, these data suggest that primary dentate gyrus and SVZ cells contain reproducible side populations that efflux DNA-binding dyes via ABC transporters, as visualized through measuring cell size together with the fluorescence of the DNA-binding dye.

### The dentate gyrus and SVZ side populations comprise CD31+ endothelial cells

We hypothesized that the side population may have endothelial cell identity, as have been previously identified in *ex vivo* early postnatal SVZ cells [3]. This hypothesis was tested using the surface marker for endothelial cells, cluster of differentiation 31 (CD31) to identify them among the dentate gyrus cell types. As shown in Fig 3, CD31 expressing (CD31+) endothelial

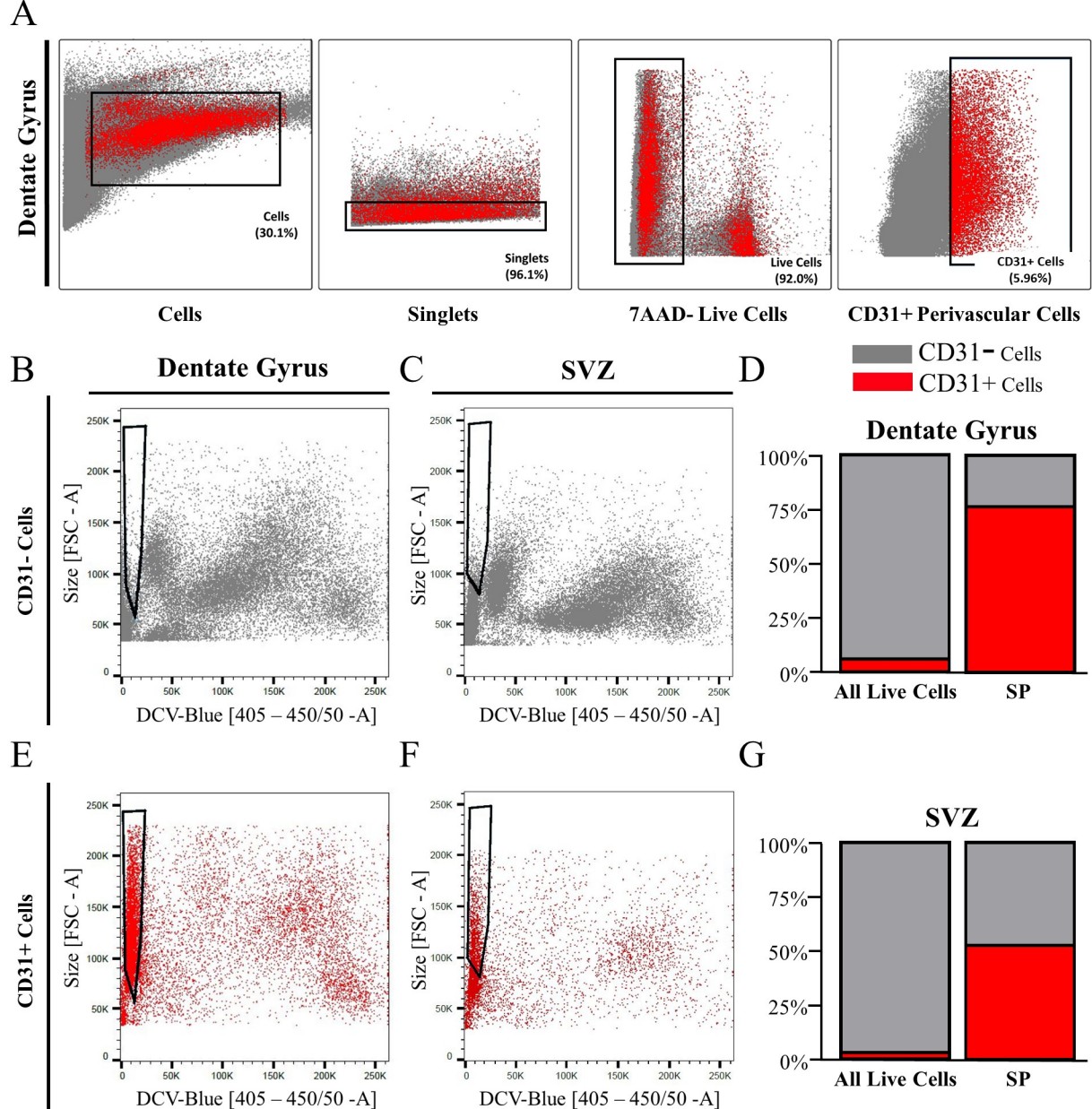

**Fig 3. The side population phenotype of adult primary dentate gyrus and SVZ cells is mainly composed of endothelial cells.** Gating of CD31+ cells (red) apart from the main population (grey) in the dentate gyrus (A) is shown as an example. The CD31-negative cells are a heterogeneous population in the dentate gyrus (B) and SVZ (C) and do not contain the DNA-dye-effluxing side population. Alternatively, CD31-positive cells localize to the side population area make up a large portion of the SP in primary dentate gyrus (D, E) and SVZ cells (F, G). Plots B and E are representative plots based on samples pooled from five male and four female mice. Plots C and F are representative plots based on samples pooled from seven male mice.

cells represented 5.96% of all cells in the dissected adult dentate gyrus (Fig 3C) and 3.07% of all cells in the dissected adult SVZ (Fig 3F). Analysis of three experiments revealed that CD31 + cells in the dentate gyrus averaged 6.35±0.37% of all live singlets and comprised 13,197 ±2,141 live single cells per sample. CD31+ cells from the SVZ were on average 3.93± 0.58% of all live singlets, and contained 6,738±974 live single cells per sample. All other cells, including the adult NSCs, were labeled as CD31-negative (CD31-) main population. CD31- cells from the dentate gyrus did not contain the majority of side population, as shown by the size-DCV-Blue plot (Fig 3B). On the contrary, CD31+ cells were heterogeneous in size and DNA content but mostly exhibited a homogenous low DCV fluorescence (Fig 3E). In fact, 75.48% of SP cells in the dentate were CD31+ cells (Fig 3D). Similarly, for the SVZ, CD31- cells from the SVZ did not contain the majority of side population cells (Fig 3C), while CD31+ cells occupied this area (Fig 3F) and represented 52.65% of the side population (Fig 3G).

Since CD31 is also a marker of hematopoietic cells, we examined whether CD31+ cells co-expressed CD45, a ubiquitous cell antigen expressed on the surface of hematopoietic cells, including monocytes [34, 35], and which has been used in flow cytometry together with CD31 to exclude blood cells [28, 36]. We found that CD45+ cells accounted for 4.80% of all live cells from the dentate gyrus (S2B Fig), and 6.43% in the SVZ (S2E Fig). The distribution of CD45 + cells among the CD31+ cells represented a small fraction of all CD31+ cells. Specifically, 5.54% of CD31+ cells in dentate gyrus were positive for CD45 (S2C Fig), and 9.87% in SVZ (S2F Fig). Thus, the fairly small overlap in CD31 and CD45 markers suggests that the vast majority of the CD31-expressing DCV-effluxing cells in the side populations of the SVZ and DG are endothelial cells rather than blood cells.

## Discussion

This series of experiments was used to examine whether a side population can be identified in primary cells of the dentate gyrus and SVZ using DCV, a live-cell-permeable DNA-binding dye, in flow cytometry. We identified a side population that effluxes DCV that was best visualized through measuring cell size together with the fluorescence of the DNA-binding dye. These cells represent the *bona fide* side population that responds to the inhibition with ABC transporter antagonists, verapamil and fumitremorgin C. In both the dentate gyrus and SVZ cells, the side population of the neurogenic regions is enriched with CD31-expressing endothelial cells.

### CD31-expressing endothelial cells are enriched within the side population of ex vivo neurogenic extracts

Our data support that within the young adult dentate gyrus and SVZ, CD31+ cells are the major cell cluster in the side population that responded to ABC transporter inhibition. Approximately 75% of dentate gyrus SP cells and 53% of SVZ SP cells were positive for CD31. More than 90% of all CD31+ cells found in the dentate gyrus and SVZ were negative for the hematopoietic marker CD45, leading us to hypothesize that these CD31+ cells are endothelial cells, and not CD31+ hematopoietic cells.

Additional support for our interpretation that the CD31+ SP cells are endothelial cells, comes from converging lines of indirect evidence from different studies. For example, data from single-cell RNA sequencing studies show few to no blood cells in *ex vivo* samples collected from naive adult dentate gyrus and SVZ tissue [19, 37, 38]. Primary adult cerebral endothelial cells show high ABC transporter protein levels [39], and single-cell RNA sequencing datasets from the human and mouse brain demonstrate that endothelial cells strongly express ABC transporter gene mRNA [40, 41]. Lastly, the results of Mouthon et al. [3] show that the

side population of early postnatal SVZ identified by Hoechst 33342 contains a majority of cells that express the endothelial cell marker CD31 and vonWillebrand factor (vWF), and do not contain NSC markers (e.g., CD133) or the pan-hematopoietic marker CD45. Together these data strongly suggest that the side population from the SVZ and dentate gyrus identified by DCV fluorescence and cell size are likely to be endothelial cells, and future studies could extend this finding when performing more extensive downstream cell analysis.

## Requirements for optimization of side population assay

This study also showed that optimization of some parameters is required for accurate side population analysis. The optimization of the DCV dilution (S1 Fig) was done in order to avoid use of low or excess concentrations of DNA-binding dyes that can lead to the false identification of low-fluorescence cells as belonging to the side population [32], as non-effluxing cells in side and main populations are often continuous. Ensuring proper excitation with optimal voltage parameters for primary brain cells (S1C and S1F Fig) is also important to capture full heterogeneity of their DNA content. In addition, the usefulness of the relative cell size parameter (FSC) cannot be understated when locating very small side populations, such as those in the dentate gyrus and SVZ. The incorporation of the ABC transporter inhibitors further allows for more precise, higher resolution identification of the *bona fide* side population. This is demonstrated by the strong evidence of efflux within the side populations of dentate gyrus and SVZ cells, with 89% and 69% cells inhibited by verapamil and fumitremorgin C, respectively, which may be due to the fact that our inhibitions do not include the ABCC family of transporters. Overall, our findings show that the dentate gyrus and SVZ side populations were reproducible and modest in size and responded to verapamil and fumitremorgin C treatment but could only be identified while gating based on DNA content together with relative cell size.

## Using SP assay to detect NSCs

The CD31+ cells represented a fairly small population of total live single cells in the dentate gyrus and SVZ, however, most of the cells within the side population area were positive for CD31. A small fraction of around 25% of dentate gyrus and 48% SVZ side population cells did not express CD31. We hypothesize that these CD31- cells are not NSCs but may be microglial cells, as previously reported in the side populations of SVZ [3], which remains to be confirmed in future work. Moreover, even if these cells are NSCs, given the majority of cells are CD31 + endothelial cells, our data shows that the side population assay would not be efficient for the isolation of NSCs. This is in direct contrast to the efficiency of the side population assay to detect NSCs from cultured embryonic and adult cells from the SVZ niche [3, 12]. Such discrepancies may point to biological differences in cultured and uncultured neurogenic cell niches, which has been suggested to be due to hypoxic conditions of neurosphere cultures [3, 12]. Independent of the cause of these differences, our findings and the work of others support that the identification of NSCs is limited to NSCs cultured *in vitro*. Additionally, we conclude that the use of DCV and analysis of the side population from *ex vivo* cell preparations from the neurogenic regions of the adult brain provides an inexpensive method to study effluxing perivascular cells.

## Supporting information

**S1 Fig. The titration of the DCV reagent in adult dentate gyrus cells.** DCV staining testing 1X (A, D) or 2X DCV (B, E, C, F), as well as varying degrees of voltage for the 2X DCV concentration (B, E vs. C, F) as shown in dual fluorescence plots (A-C) and DCV-Blue/size plots (D-F). These results suggested 2X DCV with optimal excitation (C, F) was sufficient to

distinguish the heterogeneous populations. Plots A, B, D, and E were generated from samples of pooled nine female mice. Plot C and F were generated from pooled samples of five male and three female mice, 2.5k cells are shown.
(TIF)

**S2 Fig. Characterization of CD31-expresing cells.** CD31 and CD45 staining in the dentate gyrus (A—unstained, B—all-stained) and the SVZ (D—unstained, E—all-stained) show little co-expression of CD31 and CD45 in the main populations (B and E, respectively) with only a small proportion of CD31+ cells expressing CD45 (C and F). These plots are generated based on pooled samples from five male and three female mice, 50k cells are shown.
(TIF)

**S1 File.**
(ZIP)

## Acknowledgments

We would like to thank all members of the Lagace lab for continued valued. We thank Vera A. Tang, the operations manager of the uOttawa Flow Cytometry and Virometry Core Facility for critical feedback and assistance with data collection.

## Author Contributions

**Conceptualization:** Alena Kalinina, Yingben Xue, Diane C. Lagace.

**Formal analysis:** Alena Kalinina.

**Funding acquisition:** Diane C. Lagace.

**Investigation:** Alena Kalinina, Diane C. Lagace.

**Methodology:** Alena Kalinina, Catherine Gnyra, Vera Tang, Yingben Xue, Diane C. Lagace.

**Project administration:** Alena Kalinina, Diane C. Lagace.

**Resources:** Vera Tang, Diane C. Lagace.

**Software:** Alena Kalinina.

**Supervision:** Vera Tang, Yingben Xue, Diane C. Lagace.

**Validation:** Alena Kalinina, Catherine Gnyra, Diane C. Lagace.

**Visualization:** Alena Kalinina.

**Writing – original draft:** Alena Kalinina.

**Writing – review & editing:** Alena Kalinina, Vera Tang, Yingben Xue, Diane C. Lagace.

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
