## [Decision Letter · Decision Letter 0]

27 Apr 2021

PONE-D-21-11801

Isolation of the side population from adult neurogenic niches enriches for endothelial cells

PLOS ONE

Dear Dr. Lagace,

Thank you for submitting your manuscript to PLOS ONE. After careful consideration, we feel that it has merit but does not fully meet PLOS ONE’s publication criteria as it currently stands. Therefore, we invite you to submit a revised version of the manuscript that addresses the points raised during the review process.

We look forward to receiving your revised manuscript.

Kind regards,

Marietta Zille, PhD

Academic Editor

PLOS ONE

Journal Requirements:

2)  We note that you have indicated that data from this study are available upon request. PLOS only allows data to be available upon request if there are legal or ethical restrictions on sharing data publicly. For more information on unacceptable data access restrictions, please see http://journals.plos.org/plosone/s/data-availability#loc-unacceptable-data-access-restrictions.

3) PLOS requires an ORCID iD for the corresponding author in Editorial Manager on papers submitted after December 6th, 2016. Please ensure that you have an ORCID iD and that it is validated in Editorial Manager. To do this, go to ‘Update my Information’ (in the upper left-hand corner of the main menu), and click on the Fetch/Validate link next to the ORCID field. This will take you to the ORCID site and allow you to create a new iD or authenticate a pre-existing iD in Editorial Manager. Please see the following video for instructions on linking an ORCID iD to your Editorial Manager account: https://www.youtube.com/watch?v=_xcclfuvtxQ

Reviewers' comments:

Reviewer #1: Review Kalinina et al.

In the present manuscript Kalinina et al. report a FACS based method to isolate endothelial cells from mouse brain tissue. The main purpose of the work was to isolate neural stem cells from the gyrus and subventricular zone by gating for side population characterized by low fluorescence in cells that exhibit high efflux of DNA biding dyes. Meanwhile, no neural stem cells were present in the side population, however a large proportion expresses CD31 and the authors suggest these are endothelial cells.

The manuscript is descriptive in nature and provides sound and well controlled protocol, that is relevant for peers studying endothelial cells to have access to. However, CD31 expressing cells my be of hematopoietic origin, and do not makeup the entire side population and this should be addressed in greater detail. Please find my suggestions below:

Major comments

1. A more thorough characterization of the CD31 expressing cells is warranted to draw the conclusion that they are endothelial cells. While it is true that CD31 is a marker of endothelial cells, it should also be considered that white blood cell populations such as monocytes express CD31. This could be by immunohistochemistry, RNAscope, or qPCR using genes/proteins that are specific for endothelial cells1. This would also allow for full or tentative identification of the non CD31 expressing fraction of the side population.

2. The authors should indicate if the presented plots are pooled results of samples from more animals and the abundance of cells recovered from side populations from each brain. Please also include the variability on the protocol between experimental days, i.e. are the same number cells of the side populations obtained from each brain. This is relevant for the applicability of the protocol for isolation of endothelial cells for further studies by peers. Moreover, the authors are asked to systematically report how many mice were used in each experiment (e.g. in figure text) and gender proportion of included animals.

Minor comments:

Line 55-57: Could you include an indication of why it is particularly interesting to isolate cells form this place?

Line 88: Please rephrase, the numbers are confusing. Please also consider that a C57BL/6J mouse is generally considered adult at 3 months of age2.

Line 100: degree sign must be superscript

121: Cell � Cells

165: it is a bit confusing with the figure texts in the middle of the results paragraph. But I assume this is corrected in the final version. Figure text: please indicate how many mice were used to produce these plots

Line 199: Please indicate the number of animals included in the production of these plots. And indicate the number of cells recovered in the side scatter

Line 216: The authors write the cells are heterogeneous in size and DNA content. This supports the major comment 1. That more thorough characterization is called for. Preferably fixation and imaging.

Line 217: Please include actual numbers of cell CD31 expressing cells, and number of animals included for each plot, could also be included in figure text (line 224).

274: please indicate actual populations size of side population preferably in proportion to main population.

281: Authors hypothesize that the CD31- proportion of the side population are microglia and hematopoietic cells – it would elevate the applicability of the protocol for peers significantly if the authors could provide more detailed information about this population.

Figure 3 A and F: can the authors indicate variation in respect to samples in the quantification of CD31 proportions?

1. Vanlandewijck M, He L, Mae MA, et al. A molecular atlas of cell types and zonation in the brain vasculature. Nature. 2018;554(7693):475-480.

2. Flurkey K, M. Currer J, Harrison DE. Chapter 20 - Mouse Models in Aging Research. In: Fox JG, Davisson MT, Quimby FW, Barthold SW, Newcomer CE, Smith AL, eds. The Mouse in Biomedical Research (Second Edition). Burlington: Academic Press; 2007:637-672.

Reviewer #2: This is a very interesting manuscript using FACS for identification and isolation of endothelial cells from neurogenic niches. The authors interpreted the results nicely and the conclusions are comprehensible. The proposed method for identification of endothelial cells can be of specific use if dissociation procedures use enzymes, which may cleave the epitope of interest. Overall, the study appears to be performed carefully. In my opinion, there are no major concerns about the manuscript.

Minor comments include:

- Although the existence of the BBB is discussed in the manuscript, the authors might consider discussing it earlier in the manuscript. When talking about brain ECs, the BBB is a major part. Especially when addressing ABC transporters on brain ECs.

- The authors could extend on the purity of the population when suggesting DCV staining as cheaper and easier option for EC isolation (254, 291)

- The authors mention the number of animals used for the whole study. For quantification, it could be useful to include the number of animals or the SD in e.g. the figure legend for the reader to estimate.

- Dentate gyrus, typo (229, 268)

- Figure legend for Figure 2, SVZ is (B) (199)

- Including a gating strategy to the figure would allow the reader to comprehend easily what is shown in the figure without going back to materials and methods section.

---

## [Author Response · Author response to Decision Letter 0]

28 Sep 2021

Reviewer #1: Review Kalinina et al.

In the present manuscript Kalinina et al. report a FACS based method to isolate endothelial cells from mouse brain tissue. The main purpose of the work was to isolate neural stem cells from the gyrus and subventricular zone by gating for side population characterized by low fluorescence in cells that exhibit high efflux of DNA biding dyes. Meanwhile, no neural stem cells were present in the side population, however a large proportion expresses CD31 and the authors suggest these are endothelial cells.

The manuscript is descriptive in nature and provides sound and well controlled protocol, that is relevant for peers studying endothelial cells to have access to. However, CD31 expressing cells maybe of hematopoietic origin, and do not makeup the entire side population and this should be addressed in greater detail. Please find my suggestions below:

We thank the reviewer for their time, extensive review of our manuscript, and the very helpful suggestions. 

Major comments

1. A more thorough characterization of the CD31 expressing cells is warranted to draw the conclusion that they are endothelial cells. While it is true that CD31 is a marker of endothelial cells, it should also be considered that white blood cell populations such as monocytes express CD31. This could be by immunohistochemistry, RNAscope, or qPCR using genes/proteins that are specific for endothelial cells. This would also allow for full or tentative identification of the non CD31 expressing fraction of the side population.

Thank you for this insightful comment. 

We have completed an additional experiment that supports that the CD31+ cells are unlikely to be a part of the CD45+ blood cell population. As per the attached revised manuscript and new supplemental Figure 2: 

Since CD31 is also a marker of hematopoietic cells, we examined whether CD31+ cells co-expressed CD45, a ubiquitous cell antigen expressed on the surface of hematopoietic cells, including monocytes (34,35), and which has been used in flow cytometry together with CD31 to exclude blood cells (28,36). We found that CD45+ cells accounted for 4.80% of all live cells from the dentate gyrus (Fig S2 B), and 6.43% in the SVZ (Fig S 2E). The distribution of CD45+ cells among the CD31+ cells represented a small fraction of all CD31+ cells. Specifically, 5.54% of CD31+ cells in dentate gyrus were positive for CD45 (Fig S 2C), and 9.87% in SVZ (Fig S 2F). Thus, the fairly small overlap in CD31 and CD45 markers suggests that the vast majority of the CD31-expressing DCV-effluxing cells in the side populations of the SVZ and DG are endothelial cells rather than blood cells.

We also hypothesize that our CD31 are not blood cells, due to a variety of different lines of evidence from the literature that we have cited in the revised manuscript discussion. This paragraph now reads: 

Additional support for our interpretation that the CD31+ cells are endothelial, comes from converging lines of indirect evidence from different studies. For example, data from single-cell RNA sequencing studies show few to no blood cells in ex vivo samples collected from naive adult dentate gyrus and SVZ tissue (19,37,38). Primary adult cerebral endothelial cells show high ABC transporter protein levels (39), and single-cell RNA sequencing datasets from the human and mouse brain demonstrate that endothelial cells strongly express ABC transporter gene mRNA (40,41). Lastly, the results of Mouthon et al. (3) show that the side population of early postnatal SVZ identified by Hoechst 33342 contains a majority of cells that express the endothelial cell marker CD31 and vonWillebrand factor (vWF), and do not contain NSC markers (CD133) or the pan-hematopoietic marker CD45. Together these lines of data suggest that the side population cells from the SVZ and dentate gyrus identified by DCV fluorescence and cell size are likely to be endothelial cells.

Lastly, we added a sentence within our discussion to acknowledge that despite these converging lines of support, a more downstream analysis of these cells could be done in future studies:

Together these data strongly suggest that the side population cells from the SVZ and dentate gyrus identified by DCV fluorescence and cell size are likely to be endothelial cells, that future studies could confirm when performing more extensive downstream analysis. 

2. The authors should indicate if the presented plots are pooled results of samples from more animals and the abundance of cells recovered from side populations from each brain. Please also include the variability on the protocol between experimental days, i.e. are the same number cells of the side populations obtained from each brain. This is relevant for the applicability of the protocol for isolation of endothelial cells for further studies by peers. Moreover, the authors are asked to systematically report how many mice were used in each experiment (e.g. in figure text) and gender proportion of included animals.

We apologize for not including these important methodological details in the first version of manuscript. We have included an additional note in the methods section that for each experiment, including the representative figures, cells from multiple mice were pooled into one dentate gyrus sample and one SVZ sample. In figure legends, we have included number and sex of mice used for the plots. 

In terms of variability of the proportion of the side population in different experiments, we have also included the average percentages and standard error of mean and average numbers and standard error of SP and CD31+ cells obtained from mice in different experiments in the results section: 

Examination of all performed experiments revealed that the size of dentate gyrus and SVZ side populations was reproducible between experimental days. Specifically, the average size of the dentate gyrus side population from 5 experiments was 3.82±0.26% of all live singlets, with 4050±1099 SP cells analyzed per sample. Similarly, the average size of the SVZ side population from 3 experiments was SVZ of 1.48±0.47% of all live singlets and contained 3328±1817 live single cells per sample. 

Analysis of three experiments revealed that CD31+ cells in the dentate gyrus averaged 6.35±0.37% of all live singlets and comprised 13197±2141 live single cells per sample. CD31+ cells from the SVZ were on average 3.93± 0.58% of all live singlets, and contained 6738± 974 live single cells per sample.

In the method section have also added how many cells were analyzed on average per pooled mouse samples. We did not include the percentages or cell numbers per each mouse brain since our study relied on pooled samples from multiple mice and the numbers of mice differed on experimental days. Additionally, the samples were not always run fully on the flow cytometer, therefore, it is not possible to accurately determine the total numbers of SP or CD31+ cells obtained for each mouse as also described now in the methods: 

The number of live single cells analyzed in all-stained samples averaged 160±20k live single cells for all experiments. The full entirety of the samples was not run in the experiments since it is reported that at least 25k live single cells is an optimal number for this analysis, thus other experimenters may be able to collect more live single cells from the same number of mice that is reported in this study. 

Minor comments:

Line 55-57: Could you include an indication of why it is particularly interesting to isolate cells from this place?

We have added a sentence in the introduction to highlight the importance of isolating cells from the neurogenic niches, as well as references for further reading.

Neural stem cells (NSCs) within the subventricular zone (SVZ) of the lateral ventricle and the subgranular zone of the dentate gyrus can develop into functional mature neurons in the adult brain. There is interest in harvesting cells from these regions in order to understand how NSCs and their progeny contribute to brain function in health and disease and could be harnessed for cell-based brain repair (10,11).

Line 88: Please rephrase, the numbers are confusing. Please also consider that a C57BL/6J mouse is generally considered adult at 3 months of age2.

We have rephrased the problematic wording for the number of mice

Fifty-six male and female two to three months old C57bl/6J background mice were used for all experiments.

We also modified this revised revision throughout to classify our mice are young adults, as defined in method section for 2 months of age. Additionally. we removed the term adult from title and a few other locations in manuscript.

Line 100: degree sign must be superscript

Fixed

121: Cell ? Cells

Fixed

165: it is a bit confusing with the figure texts in the middle of the results paragraph. But I assume this is corrected in the final version

We completely agree the location of the figure legends in the text are confusing and were surprised by to read the PLOSone journal guidelines that: the “Figure captions must be inserted in the text of the manuscript, immediately following the paragraph in which the figure is first cited (read order). Do not include captions as part of the figure files themselves or submit them in a separate document.” 

Therefore, as requested by reviewer we have moved the figure legends to all appear after discussion. This can be adjusted if accepted for publication and required by PLOSOne.

p165. Figure text: please indicate how many mice were used to produce these plots

Line 199: Please indicate the number of animals included in the production of these plots. And indicate the number of cells recovered in the side scatter

We have included the number of animals used within figure legends to produce the plots.

Since we have used the singlet and live cell discrimination in addition to debris in side scatter, we included the average number of live single cells recovered for each sample in the Methods under flow cytometry, as it better represents the usable cells recovered from a sample. 

The number of live single cells analyzed in all-stained samples averaged 160±20k live single cells for all experiments.

Line 216: The authors write the cells are heterogeneous in size and DNA content. This supports the major comment 1. That more thorough characterization is called for. Preferably fixation and imaging.

As reviewed above in major comment #1, we have strengthened our interpretation that CD31+ cell are endothelial cells, and not blood cells by performing an additional experiment and citing of additional data. The heterogeneity in the size and DNA content is not unexpected and we hypothesize this occurs due to different causes including 1) variability of DNA content and cell size during cell cycle phases and response to microenvironmental cues; 2) the growing appreciation for the diversity in the population of different endothelial cell subtypes (13–15); and 3) the difference in the DNA content of the SP CD31+ cells expelling the DNA dye compared to main population CD31+ cells.

Line 217: Please include actual numbers of cell CD31 expressing cells, and number of animals included for each plot, could also be included in figure text (line 224).

Animal numbers and sex proportions were added to all the figure legends. 

We have included actual numbers of CD31+ cells within the results:

Analysis of three experiments revealed that CD31+ cells in the dentate gyrus averaged 6.35±0.37% of all live singlets and comprised 13197±2141 live single cells per sample. CD31+ cells from the SVZ were on average 3.93± 0.58% of all live singlets, and contained 6738± 974 live single cells per sample.

274: please indicate actual populations size of side population preferably in proportion to main population.

We added the size of the inhibited side population as percentage of total side population in the results section:

This is demonstrated by the strong evidence of efflux within the side populations of dentate gyrus and SVZ cells, with 89% and 69% cells inhibited by verapamil and fumitremorgin C, respectively

281: Authors hypothesize that the CD31- proportion of the side population are microglia and hematopoietic cells – it would elevate the applicability of the protocol for peers significantly if the authors could provide more detailed information about this population.

As noted in major comment 1 we have added an additional experiment and literature that suggests hematopoietic cells are unlikely to be in the side population, and thus revised this statement in the discussion to remove the suggestion that they are hematopoietic cells. This section of the discussion is revised as:

The CD31+ cells represented a fairly small population of total live single cells in the dentate gyrus and SVZ, however, most of the cells within the side population area were positive for CD31. A small fraction of around 25% of dentate gyrus and 48% SVZ side population cells did not express CD31. We hypothesize that these CD31- cells are not NSCs but may be microglial cells, as previously reported in the side populations of SVZ (3), which remains to be confirmed in future work.

Figure 3 A and F: can the authors indicate variation in respect to samples in the quantification of CD31 proportions?

As per previous comment, we have added this information in the results section: 

Analysis of three experiments revealed that CD31+ cells in the dentate gyrus averaged 6.35±0.37% of all live singlets and comprised 13197±2141 live single cells per sample. CD31+ cells from the SVZ were on average 3.93± 0.58% of all live singlets, and contained 6738± 974 live single cells per sample.

Reviewer #2: 

This is a very interesting manuscript using FACS for identification and isolation of endothelial cells from neurogenic niches. The authors interpreted the results nicely and the conclusions are comprehensible. The proposed method for identification of endothelial cells can be of specific use if dissociation procedures use enzymes, which may cleave the epitope of interest. Overall, the study appears to be performed carefully. In my opinion, there are no major concerns about the manuscript.

We appreciate the time reviewer #2 put into reviewing our paper and the helpful minor suggestions for changes. 

Minor comments include:

- Although the existence of the BBB is discussed in the manuscript, the authors might consider discussing it earlier in the manuscript. When talking about brain ECs, the BBB is a major part. Especially when addressing ABC transporters on brain ECs.

As suggested we have added the following to the introduction:

This finding is not surprising, since endothelial and microglial cells, along with pericytes and astrocytes form and maintain the blood brain barrier (16–18). Accordingly, one of the main roles of endothelial cells is in brain homeostasis, which relies on the function of the ABC transporters (18). 

- The authors could extend on the purity of the population when suggesting DCV staining as cheaper and easier option for EC isolation (254, 291)

As similarly requested by review #1 major comment #1 and described above we have performed additional experiments with CD45, that revealed more than 90% of CD31+ cells in the dentate gyrus and the SVZ are not hematopoietic. 

- The authors mention the number of animals used for the whole study. For quantification, it could be useful to include the number of animals or the SD in e.g. the figure legend for the reader to estimate.

We have provided the number and gender of animals in the figure legends, as requested also by Reviewer #1 in comment 2.

- Dentate gyrus, typo (229, 268) 

Fixed.

- Figure legend for Figure 2, SVZ is (B) (199)

Fixed.

- Including a gating strategy to the figure would allow the reader to comprehend easily what is shown in the figure without going back to materials and methods section.

We have added a gating strategy image to Figure 3.

---

## [Decision Letter · Decision Letter 1]

6 Dec 2021

Isolation of the side population from neurogenic niches enriches for endothelial cells

PONE-D-21-11801R1

Dear Dr. Lagace,

We’re pleased to inform you that your manuscript has been judged scientifically suitable for publication and will be formally accepted for publication once it meets all outstanding technical requirements.

Kind regards,

Marietta Zille, PhD

Academic Editor

PLOS ONE

Additional Editor Comments (optional):

All comments have been addressed.

---

## [Editor Report · Acceptance letter]

13 Dec 2021

PONE-D-21-11801R1 

Isolation of the side population from neurogenic niches enriches for endothelial cells 

Dear Dr. Lagace:

I'm pleased to inform you that your manuscript has been deemed suitable for publication in PLOS ONE. Congratulations! Your manuscript is now with our production department. 

Kind regards, 

on behalf of

Prof. Dr. Marietta Zille 

Academic Editor

PLOS ONE